# Experimental In Vitro Microfluidic Calorimetric Chip Data towards the Early Detection of Infection on Implant Surfaces

**DOI:** 10.3390/s24031019

**Published:** 2024-02-05

**Authors:** Signe L. K. Vehusheia, Cosmin I. Roman, Markus Arnoldini, Christofer Hierold

**Affiliations:** 1Department of Mechanical and Process Engineering, ETH Zurich, 8092 Zurich, Switzerland; cosmin.roman@micro.mavt.ethz.ch (C.I.R.); christofer.hierold@micro.mavt.ethz.ch (C.H.); 2Department of Health Sciences and Technology, ETH Zurich, 8093 Zurich, Switzerland; markus.arnoldini@hest.ethz.ch

**Keywords:** heat flux measurement, microfluidics, microbiology

## Abstract

Heat flux measurement shows potential for the early detection of infectious growth. Our research is motivated by the possibility of using heat flux sensors for the early detection of infection on aortic vascular grafts by measuring the onset of bacterial growth. Applying heat flux measurement as an infectious marker on implant surfaces is yet to be experimentally explored. We have previously shown the measurement of the exponential growth curve of a bacterial population in a thermally stabilized laboratory environment. In this work, we further explore the limits of the microcalorimetric measurements via heat flux sensors in a microfluidic chip in a thermally fluctuating environment.

## 1. Introduction

Heat flux measurements are very interesting for measuring the metabolic heat output of chemical and biological systems [1]. For example, heat produced by living organisms reflects metabolic activity and metabolic changes [2,3]. Large instruments such as microcalorimeters have been used for studying the heat produced during the growth of different bacterial strains in medicine [4] and microbiology [5,6]. On-chip calorimetric measurements such as micro- [7,8], nano- [9,10,11] and pico-calorimetric chips [12] have also been utilized for investigating metabolic changes through exothermic heat production. Microfluidic systems offer a number of benefits, such as optical access, controllable fluid mixing and lab-on-chip/in vitro systems capabilities, as summarized in the literature [13]. Many different microfluidic chips have been developed to emulate the in vivo environment in the body for diagnostic [14,15] or research purposes of different regions in the body such as the kidney [16], lung [17] or tumors [18]. These systems focus on the detection of biochemical markers or on optical investigations of the in vitro system. We wish to adapt the concept of in vitro microfluidic systems to emulate the in vivo environment of an infection and use physical sensors as opposed to biochemical sensors to avoid drift and promote long-term stability of the sensor in vivo. Having heat flux measurements accessible in microfluidic chips would expand the range of tools for bioengineering with tabletop access to microcalorimetric data. This work is an extension of work presented at the XXXV Eurosensors Conference 2023 in Lecce (10–13 September).

Microcalorimetric microfluidic chips have been proposed to mimic the thermal environment of a growing infection on a vascular graft implant surface [19,20]. For example, Figure 1a shows a 2 + 1 channel microfluidic system, where a growing infection is emulated by an exothermic chemical reaction. The heat produced is measured differentially by two heat flux sensors. Heat transfer to blood flowing in an aorta is accounted for by a top channel. Its heat transfer coefficient, h, is designed to match that of a physiological aorta [19].

In another system, shown in Figure 1b, the exponential growth curve of a bacterial population was measured using a similar concept involving differential heat flux sensors [1] in a microfluidic chip. The individual bacterial thermal power could be determined thanks to a thermally stabilized environment. This system was developed to facilitate calorimetric measurements on microfluidic chips in laboratory environments. The thermally stabilized environment consists of an incubator at 37 °C, a PMMA (Poly(methyl methacrylate)) thermally stabilizing box and a block of copper placed on top of the two channels to keep the thermal fluctuation to a minimum.

These in vitro experimental systems have shown the potential application of heat flux sensing towards the early detection of infection. However, some aspects still need to be investigated before considering in vivo applications. On implant surfaces, to detect an infection early, one has to resolve the onset of bacterial growth prior to biofilm formation, after which bacteria stop responding to antibiotic treatment. Such early detection could significantly decrease the patient mortality rate of, for example, vascular graft implant infections, which is currently between 17 and 40% of cases [21]. In the system shown in Figure 1b, the resolution with respect to the number of bacteria is supported by a thermally stabilized environment [1]. In contrast, in vivo, the thermal environment of the implant would undergo temperature fluctuations (sleep cycles, outside temperature, level of patient activity, and fever, for example). To determine the feasibility of the early detection of infection in vivo, it is thus important to investigate the influence of higher temperature fluctuations on the measurement capabilities of microcalorimetric chip systems. Furthermore, smaller heat flux sensors than those used in [1,19] would be needed for integration of the heat flux sensor in an aortic vascular graft, for example in a mesh structure [20]. Since smaller sensors might be more limited by noise, their impact on the resolution of measuring bacterial growth needs to be studied.

Here, we present a microfluidic chip measuring bacterial growth through heat flux in a thermally not-stabilized environment. The system uses smaller heat flux sensors and a smaller microfluidic channel (smaller thermal mass). We investigate the system in a fluctuating temperature range of 1 K, which matches the standard body temperature fluctuations in the human body (36.5–37.5 °C) [22]. This study gives further information about the limit of detection with respect to bacterial growth possible on implant surfaces, a step closer towards in vivo conditions, and extends the study presented at the Eurosensors Conference.

## 2. Materials and Methods

### 2.1. Experimental System and Setup

Two different experimental systems, as illustrated in Figure 2, will be compared. To the left, Figure 2a,c,e show the microfluidic chip in the thermally not-stabilized environment, and to the right, Figure 2b,d,f show the thermally stabilized system as introduced in [1]. Pictures of both experimental systems are shown in the Appendix A.

The two channels shown in Figure 2a,c,e are 170 μm tall, 6 × 6 mm wide and long with a volume of 6 μL. The heat flux sensor is a gSKIN XM (greenTEG, Zurich, Switzerland, resolution: 0.41 W/m^2^) with dimensions measuring 5 mm × 5 mm × 0.5 mm, whereas the two channels shown in Figure 2b,d,f are 320 μm tall and 12 × 12 mm wide and long (volume = 46 μL) with a heat flux sensor of 10 mm × 10 mm and 0.5 thickness [1]. The heat flux sensor used is the gSKIN XP (resolution: 0.06 W/m^2^). In both systems, one of the channels is filled with bacteria and the other one is filled with sterile lysogeny broth (LB) media for differentially compensated heat flux measurements. The heat flux sensors are embedded in polydimethylsiloxane (PDMS) and placed 150 μm above the channels.

Both the thermally not-stabilized setup and the thermally stabilized setup are shown comparatively in Figure 3. In the former, the microfluidic chip is placed in a temperature-controlled room at 37 ± 0.5 °C without any further temperature stabilization. A PT1000 temperature sensor measures the temperature in close proximity to the microfluidic chip. In the thermally stabilized system [1], the microfluidic chip is placed within a double-walled PMMA box with an additional thermally insulating air layer, as illustrated in Figure 3. The PMMA box is placed inside an incubator at 37 ± 0.15 °C. As also shown in Figure 2b,d,f, a copper block is placed directly above the microfluidic chip for further thermal stability of the heat flux sensors.

*Escherichia coli* (*E. coli*) MG 1655 in lysogeny broth (LB) is utilized for bacterial experiments. The LB medium consists of 10 g of Tryptone, 5 g of yeast extract, and 5 g of NaCl per 1 L of media. To prevent bacterial cells sticking to the tubing walls or in the channel, a 1:100 ratio of Tween20 is added to the LB medium. Each experiment is performed in two steps, with an initial calibration phase in a sterile environment. First, the flow is controlled using a peristaltic pump connected to both channels of the microfluidic chip. LB without bacteria is pumped at a flow rate of 98.75 μL/min. Second, bacteria are added to one of the channels, into the flask connected to the respective channel. PTFE tubing of an inner diameter of 0.02 in is inserted in the microfluidic chip inlet and outlet and connected to the peristaltic pump. The growth of the bacteria is measured via optical density (OD) analysis of the bacterial population at the outlet of the tubing from the microfluidic chip. The samples collected at the outlet are put on ice and measured subsequently using an optical density (OD) meter. This experimental process is based on a previously published protocol of the thermally stabilized system [1]. During the setup and the initial filling of the channels with liquid, it is important to ensure that no air bubbles are present in the microfluidic channels prior to the start of the measurement.

### 2.2. Microfluidic Chip Fabrication Steps

The lithography and soft lithography microfluidic chip fabrication steps are illustrated step-by-step in Figure 4. The microfluidic chip is fabricated by patterning SU8 on a silicon wafer. A 4 mL quantity of SU8-100 is spin coated on the wafer, with parameters as described in the Appendix A, for a total thickness of 170 μm. The wafer is soft baked at 95 °C for 70 min and subsequently exposed with 400 mW/cm^2^ on a mask aligner (MA6 Karl Suss, Suss Microtec SE, Garching, Germany). The mask used for the exposure contains the dimensions of the two channels, as shown in Figure 2. Following the exposure, the SU8 is baked at 95 °C for 16 min and subsequently developed in Mr Dev600 (micro resist technology GmbH, Berlin, Germany) for 16 min. After cleaning the wafer with isopropyl alcohol and DI water, it is ready for dicing and the soft lithography steps with PDMS.

PDMS (Sylgard 184, Suter-Kunststoffe AG, Fraubrunnen, Switzerland) is mixed in a 1:10 ratio between the elastomer and crosslinker and poured on the prepared and diced silicon wafer with the desired channel diameters fabricated on top, as shown in Figure 4. Following 40 min of degassing in a vacuum desiccator, the PDMS is cured at 80 °C for 3 h. Spacers are used to define the 150 μm PDMS thickness above the microfluidic channels, as shown in the sketch. Following the first layer of PDMS, the two heat flux sensors are placed on top of the cured PDMS and above the SU8 structures. More uncured PDMS is poured on top of the sensors and degassed for 40 min. The PDMS is again cured at 80 °C for 3 h.

The PDMS with the embedded heat flux sensors is cut out from the SU8 patterned wafer mold. Inlet and outlets are punched out of the PDMS using a 20 G needle. The PDMS is then fused with a glass slide using oxygen plasma ashing for 1 min and 30 s at 100 W in an O_2_ environment. For further adhesion improvement, the fused chip is directly placed in an oven at 80 °C for 15 min.

The fabrication steps are different to the fabrication steps of the microfluidic chip used in the thermally stabilized environment. The channel height for that system is 320 μm, and the specific fabrication steps are described in detail elsewhere [1].

## 3. Results


*Temperature Measurement*


Temperature fluctuations of the environment of both the thermally not-stabilized and thermally stabilized environment, as measured by the PT1000 temperature sensors, are shown in Figure 5. The thermally not-stabilized system shows a temperature fluctuation range of about 1 K for 2.3 h, and the thermally stabilized system shows a temperature fluctuation range of 0.3 K for over 4.2 h. The different temperature fluctuation ranges are due to differences in thermal shielding of the microfluidic chip from the environment, as described in the previous sections.

As shown in Figure 2a, two microfluidic channels with different functions are used. One contains only LB media for control purposes, and the other contains LB media together with growing *E. coli*. In the calibration phase, parameters for the differential compensation scheme are determined. These are thereafter applied to heat flux signals in the sensing phase when bacteria are added. The differential compensation scheme allows for the extraction of heat produced by the bacterial growth via a common-mode rejection scheme, as described previously [1]. Figure 6 shows the heat flux signal of the two sensors in the calibration phase (Figure 6a,b) and upon the addition of *E. coli*.

Figure 6a shows raw and averaged data for the calibration phase where both channels are filled with LB media only. The differentially compensated heat flux signal is shown in Figure 6b and is calculated according to the scheme introduced in [1]:(1)$$q=q_{1}-q_{2}^{*}$$

With q as the differentially compensated heat flux signal, $q_{1}\;$as the heat flux in the channel in which bacteria is to be added, $q_{2}$ as the heat flux in the control channel, $\bar{q}\;\mathrm{indicates}\;\mathrm{the}\;\mathrm{average}$ over 200 datapoints, $L_{c}$ is the ratio of the standard deviations $\sigma_{c1}/\sigma_{c2}$, and $q_{c1}\;\mathrm{and}\;q_{c2}\;\mathrm{indicate}\;\mathrm{the}\;$calibration phase [1]. $q_{2}^{*}$ is defined as
(2)$$q_{2}^{*}=L_{c}\left(q_{2}-{\bar{q}}_{c2}\right)+{\bar{q}}_{c1}$$

The differential compensation scheme manages to cancel out temperature peaks in the heat flux upon thermal fluctuations [1]. Figure 6c shows the data of the different channels upon addition of *E. coli* in one of the channels. The blue signal is from the heat flux sensor above the channel with bacteria, and the red signal is from the heat flux sensor above the calibration channel. Equations (1) and (2) yield the differentially compensated heat flux value, which are also used upon the addition of bacteria, as shown in Figure 6d. At around 8000 s, the heat flux signal increases like a step-function, as indicated by the arrow in Figure 6c,d.

## 4. Discussion

Figure 7 compares the differentially compensated heat flux signals in the thermally not-stabilized and the thermally stabilized system. Figure 7a,d show the measured heat flux data in parallel with the optical density measurement of the bacterial population. Data for the thermally not-stabilized system shown in Figure 7a is only able to reveal a step-function-like increase in the heat flux. On the other hand, heat flux measurements in the thermally stabilized setup in Figure 7d is able to show a clear exponential growth of the bacteria with the heat flux measurement. In our previous work [1], we have shown that the exponential region of the heat flux measurement matches the exponential region as measured in the optical density measurement. The larger noise in the heat flux values around the averaged signal visible in Figure 7a,c, reflects the two different models of heat flux sensors used.

To distinguish the regions in which the heat flux produced by the bacteria is below or above the detection limit for the thermally not-stabilized system, empirical cumulative distribution functions (eCDF) are utilized in different regions. The first is the calibration phase shown in Figure 6b. The second is Region 1, as indicated in Figure 7a by a horizontal line from *t* = 0 to 7500 s. The third is Region 2, as indicated by the horizontal line from *t* = 7500 to 13,000 s. Comparing eCDFs of these three regions in Figure 7b, the calibration phase and Region 1 are very close. Region 2 is significantly different from both the calibration and Region 1. There is a clear shift in the eCDF at the point where the heat produced by the bacteria surpasses the threshold of detection of the heat flux sensor. The median of the heat flux difference in the calibration phase is 0.01 W/m^2^; in Region 1 it is 0.00 W/m^2^, and in Region 2 it is 0.35 W/m^2^, as indicated by the horizontal lines in Figure 7a. The detection of bacterial growth in the thermally not-stabilized system occurs at an optical density of 0.3. In comparison, the system with the larger, more sensitive sensor [1] is able to resolve the exponential growth of the bacteria in correlation to the optical density measurement.

Table 1 compares the performance of the thermally not-stabilized and the thermally stabilized systems. The differential measurement scheme present in both systems measures metabolic heat flux and suppresses the influence of large temperature peaks and fluctuations. The spread in heat flux values are mainly related to the resolution limitations of the heat flux sensors themselves. The different detection limits are calculated via an optical density conversion, where the optical density unit corresponds to 10^9^ cells/mL. The detection limit of the cells is the calculated number of cells at the determined heat flux limit of detection converted using the given channel volume (6 μL or 46 μL). We show in Figure 7b that it is still possible to detect bacterial growth with smaller sensors and smaller thermal volumes.

## 5. Conclusions

This work extends our study presented at the Eurosensors conference. Here, we are able to resolve bacterial growth via the metabolic heat produced in a microfluidic channel with an integrated differential heat flux sensing system by measuring the bacterial growth via metabolic heat. In contrast to the previous system involving more controlled thermal environments and larger sensors and microfluidic channels [1], we expose a smaller microfluidic chip with a smaller size and worse-resolution sensors to a thermally fluctuating environment without additional thermal isolation around the microfluidic chip. The 1 K range of temperature fluctuations matches the in vivo temperature fluctuation range in the human body [22]. However, the rate of change of temperature is within minutes, which is faster than in physiological conditions. These higher rates of temperature change perturb the heat flux measurement of the bacterial growth more than what is expected in physiological conditions. We demonstrate that, leveraging a differential sensing strategy, our system can detect bacterial growth in the exponential growth phase (OD = 0.3), despite the large environmental temperature fluctuations, smaller sensor sizes, and different sensor performances.

Further, assuming that each *E. coli* bacteria produces 3.5 pW [23] of thermal power, we estimate the limit of detection to be 1707 W/m^3^ for the thermally stabilized system [1]. The limit of detection is determined by extrapolation based on sensitivity and noise. This method cannot be applied for the thermally not-stabilized system, where the limit of detection in relation to the bacterial population is determined from the OD level that causes a detecable increase in the heat flux measurement using the cumulative distribution function (Figure 7b). Therefore, we refrain from calculating the heat flux limit of detection in this case.

The bacterial concentration which constitutes the early onset of infectious growth on implant surfaces is at present unknown. Therefore, we compare the experimentally determined heat flux limit of detection to the thermal density of a biofilm of 350,000 W/m^3^ [1]. The limit of detection in our system corresponds to 0.3% of the thermal density of a biofilm. Whether or not 0.3% of the heat of a biofilm corresponds to the early stages of biofilm formation is yet to be investigated in a clinically-relevant environment. In the literature, there are papers investigating implant infections which use implant bacterial seeding amounts between 2.0 × 10^6^ and 6.2 × 10^6^ [24,25,26], which is comparable to the determined detection limit in this study. From this point of view, using bacterial growth in LB media only reflects the total heat produced by a given bacterial population, which we, due to the low bacterial population at the determined limit of detection, consider being at the onset of infection.

We thus demonstrate the feasibility of bacterial growth detection using the cumulative distribution function. The detection of bacterial growth is statistically confirmed by comparing the cumulative distribution functions of the calibration phase to the point at which the bacterial growth is detected. Admittedly, this scheme is only able to detect whether a bacterial colony has grown beyond a threshold and is not able to follow the actual exponential growth curve of the bacterial colony.

These setups prove the possibility of introducing microcalorimetric measurements on microfluidic chips in thermally fluctuating environments. They may find use as tabletop microfluidic systems for more accessible microcalorimetric measurements of samples in microfluidic chips. Furthermore, these results give an additional insight in the possibility of using heat flux sensors in thermally not-stabilized environments such as implant surfaces. Further investigations should emulate in vivo environments and investigate the feasibility of surface-covering infection detection on implants, for example via a mesh of sensors on the implant [20]. We show that by applying the differential compensation scheme, the heat flux signal removes the influence of the thermal background fluctuations in the environment around the area of interest. The smaller size of the sensor could allow for integration, for example, in the vascular graft implant wall material, and could open the way to bacterial growth detection directly at the vascular graft implant surface.

## Figures and Tables

**Figure 1 sensors-24-01019-f001:**
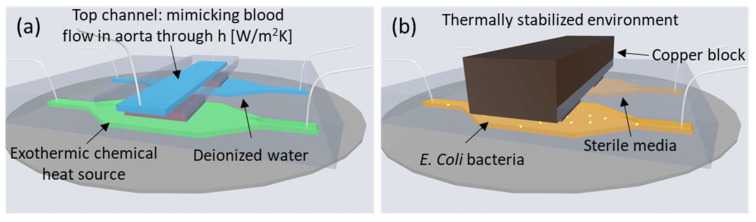
(**a**) Sketch of a 2 + 1 channel microfluidic chip [19] reproduced with permission. The heat transfer coefficient of the top channel matches that of aortic heat transfer. (**b**) Microfluidic chip in a thermally stabilized environment able to resolve bacterial growth though heat flux measurement [1]. Schematics adapted from [1,19].

**Figure 2 sensors-24-01019-f002:**
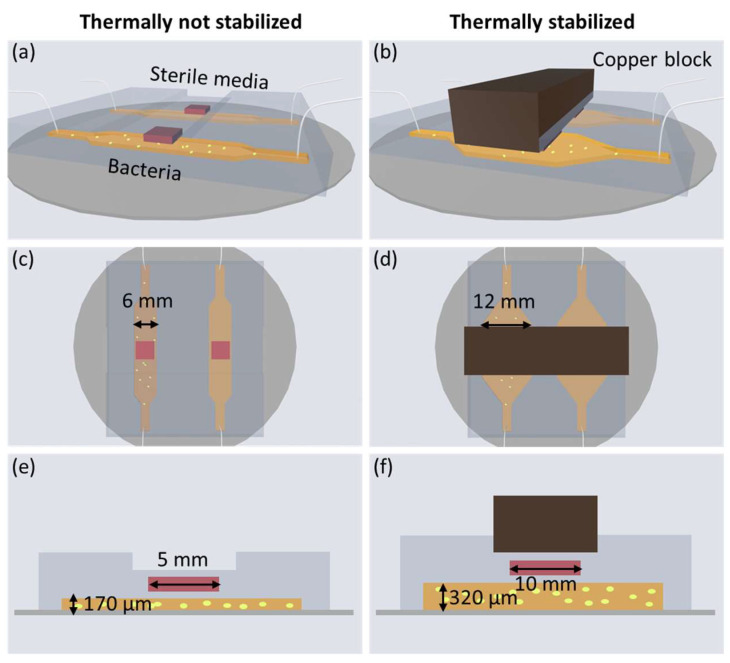
Left column (**a**,**c**,**e**) show the channels of the microfluidic chip in the thermally not-stabilized environment. Right column (**b**,**d**,**f**) show the channels of the microfluidic chip in the thermally stabilized environment [1] with the additional thermally stabilizing copper block. (**e**,**f**) show the cross-section of the chip. The groove is to ensure a given thickness of the PDMS above the heat flux sensor. Sketches not to scale.

**Figure 3 sensors-24-01019-f003:**
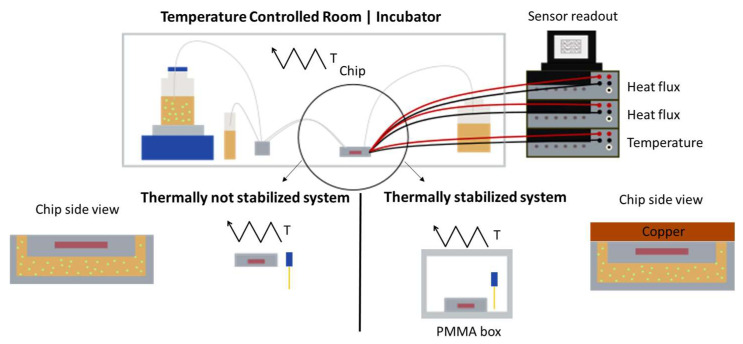
A schematic of the measurement setups of the thermally not-stabilized and the thermally stabilized systems. In the thermally not-stabilized system, the microfluidic chip is placed directly on an arbitrary surface in a temperature-controlled room without further thermal stabilization measures. The thermally stabilized system is a previously published system which shows the data of a microfluidic chip placed inside a thermally stabilizing PMMA box and with a block of copper on the top side of the microfluidic chip [1].

**Figure 4 sensors-24-01019-f004:**
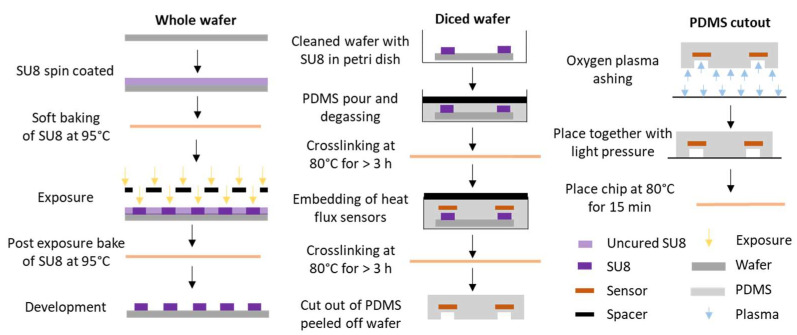
Fabrication steps of the microfluidic chip with two integrated heat flux sensors. A patterned silicon wafer with SU8 is used as the basis of the microfluidic chip channel fabrication. Multiple steps of PDMS curing are applied to create and control different layer thicknesses around the heat flux sensors. After removing the cured PDMS from the wafer, the inlets and outlets are punched out, and it is fused with a glass slide using oxygen plasma.

**Figure 5 sensors-24-01019-f005:**
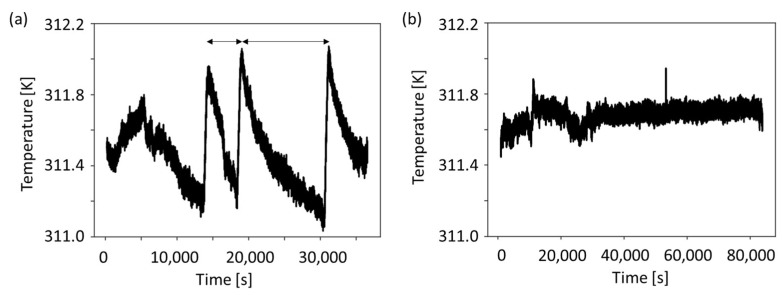
Temperature measurements in both the (**a**) thermally not-stabilized and (**b**) thermally stable environments. The fluctuations in (**a**) are in the range of 1 K from 1.0 h to 3.3 h, whereas in (**b**) the range of temperature is 0.3 K over 4.2 h. The data shown is for the duration of the whole experiment—both calibration and bacterial phase. The temperature data shown in (**b**) is part of the previously published heat flux data [1].

**Figure 6 sensors-24-01019-f006:**
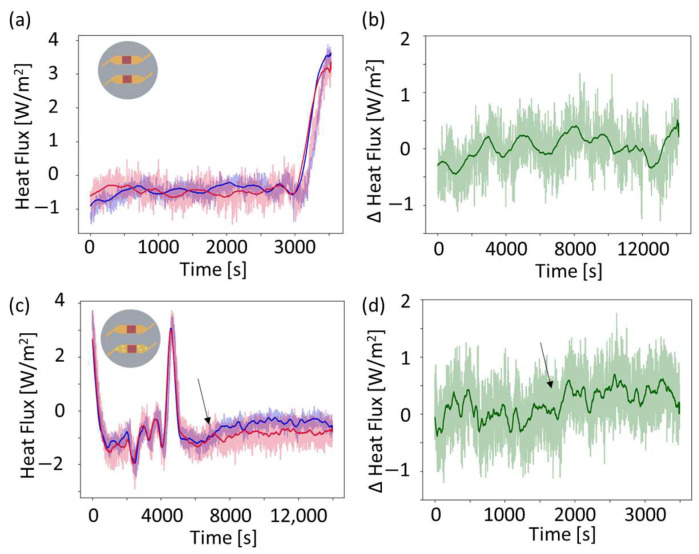
Raw and differentially compensated heat flux values. (**a**) Data during the calibration phase and (**b**) data after the differential compensation is applied. The peaks in the heat flux correspond to peaks in the temperature of the temperature-controlled room, as visible in Figure 5a. (**a**,**c**) Both have a blue heat flux signal and a red heat flux signal where the different heat fluxes represent the different channels. The blue data belong to the channel in which the bacteria is subsequently added, and the red data belong to the calibration channel. (**c**,**d**) Heat flux data upon addition of *E. coli*. Peaks in the heat flux correspond to peaks in the temperature, as visible in Figure 5a (shifted by time). The raw data are shown as opaque, and the data averaged over 200 datapoints are shown in the darker color and with a thicker line.

**Figure 7 sensors-24-01019-f007:**
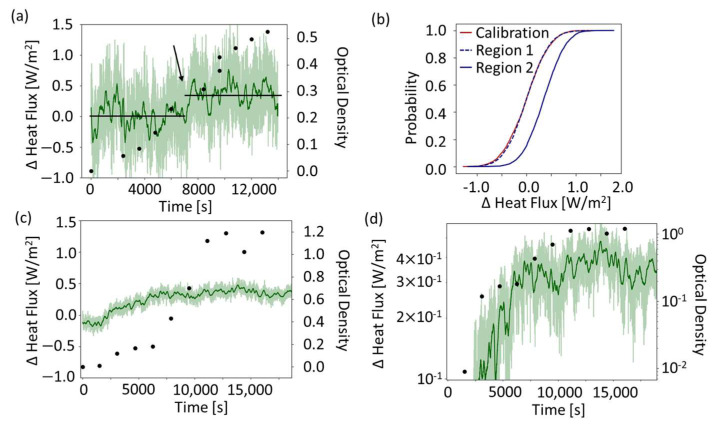
Differentially compensated heat flux measurements in the two systems. (**a**) Data in the thermally not-stabilized system. A change in the heat flux is distinguishable in the exponential growth phase (as indicated by the arrow at 7500 s). Region 1 is indicated by a line in the first part before the arrow, and Region 2 is indicated by the line starting around 7500 s. (**b**) Comparison of the cumulative distribution functions of the calibration phase with raw data with those of Regions 1 and 2. (**c**) Data for the thermally stabilized system, shown previously in [1]. The data shown are the points at which the bacteria was added to the system (*t* = 0 s is the addition of bacteria). In all figures, the opaque data is the raw heat flux signal, and the dark line shows a 200-point moving average. (**d**) Semilogarithmic plot of the data. An exponential increase in both the measured heat flux and also the bacterial population growth is identifiable in the same region.

**Table 1 sensors-24-01019-t001:** Table overview over system performance and design properties of both the thermally not-stabilized and the thermally stabilized system [1].

Property	Thermally Not-Stabilized	Thermally Stabilized [1]
Growth detection	Yes	Yes
Channel size (underneath sensor)	6 mm × 6 mm × 170 μm	12 mm × 12 mm × 320 μm
Channel volume	6 μL	46 μL
Sensor	gSKIN XM	gSKIN XP
Sensor resolution	0.41 W/m^2^	0.06 W/m^2^
Temperature fluctuation	1 K	0.3 K
Standard deviation heat flux	0.32 W/m^2^	0.05 W/m^2^
Standard deviation averaged heat flux	0.20 W/m^2^	0.02 W/m^2^
OD limit of detection *	3 × 10^8^ cells/mL	2 × 10^7^ cells/mL
Cell population limit of detection *	1.8 × 10^6^ cells	9.2 × 10^5^ cells

* These values are given for completeness; however, they are determined by different methods, and therefore direct comparison between the systems should be carefully considered.

## Data Availability

The data is available upon request to the corresponding author.

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
