# Peer review of "Experimental In Vitro Microfluidic Calorimetric Chip Data towards the Early Detection of Infection on Implant Surfaces"

_sensors, 2024, doi:10.3390/s24031019_

Round 1

Reviewer 1 Report

Comments and Suggestions for Authors

The study investigates the use of heat flux sensors for early detection of infectious growth, particularly on implant surfaces such as aortic vascular grafts. Two experimental systems are compared: a thermally not stabilized environment and a thermally stabilized system. The differential measurement scheme in both systems measures metabolic heat flux and suppresses the influence of large temperature peaks and fluctuations. The study demonstrates the potential of using heat flux sensors to detect bacterial growth and explores the limits of microcalorimetric measurement for early infection detection. The research contributes to the field of cellular metabolic activity and metabolic changes by showing the correlation between heat flux measurements and bacterial population growth. The findings suggest that it is possible to detect bacterial growth with smaller sensors and smaller thermal volumes. The study provides valuable insights into the early detection of infection on implant surfaces and opens avenues for further exploration in this area. I have the following comments and questions that authors may find useful.

The study presents an innovative approach to using heat flux sensors for early detection of infectious growth, particularly on implant surfaces.

How can the study be modified to address the limitations of the current experimental systems, such as the need for a thermally stabilized environment?

What are the potential clinical implications of the study's findings for improving patient outcomes and infection management?

How might the small sample size in the study affect the generalizability and robustness of the results, and what steps could be taken to address this limitation?

What are the potential challenges or barriers to translating the study's findings into real-world clinical practice, and how might these impact the feasibility of implementing heat flux sensors for early infection detection?

Authors should expand the introduction with more detailed criticism of the existing literature. 

Reviewer 2 Report

Comments and Suggestions for Authors

The article proposed a microfluidic chip with smaller heat flux sensors and a smaller microfluidic channel to measure bacterial growth in a not stabilized thermal environment. The article can be considered for publication in " Sensors " after revising the following questions. The comments are below.

1) In the introduction, the authors explained on-chip calorimetric measurements and advantages of microfluidic chips. However, this part of the content lacks the support of references. These several articles introducing microfluidic technology are worth learning.

Journal of Materials Chemistry B, 2023, 11, 1978-1986. DOI: 10.1039/d2tb02338e

Lab on a chip, 2017, 17, 2225. DOI: 10.1039/c7lc00249a

Biosensors and Bioelectronics, 2023, 230, 115586. DOI: 10.1016/j.bios.2023.115586

Small Methods, 2019, 4(4), 1900451. DOI: 10.1002/smtd.201900451

Biosensors and Bioelectronics, 2018, 121, 272-280. DOI: 10.1016/j.bios.2018.08.061

2) After reading the full text, we understand that the author has conducted several experiments with the system and provided a wealth of data. We are interested in the physical version of the detection system used by the authors. Has the author manufactured the system based on Figure 3? If so, please provide the real picture of the detection system.

3) The accuracy of the position of the heat flux sensor is an important factor affecting the experimental results. It is significant to provide photos of the microfluidic chip to assess its manufacturing accuracy.

4) The authors believe that this work can be used to detect bacterial infection on implant surface. However, the system in this paper detects the bacterial concentrations in LB medium. This is not consistent with the surface environment of the implant surface, and the author needs to explain this problem.

Round 2

Reviewer 1 Report

Comments and Suggestions for Authors

The authors revised the manuscript according to my comments and suggestions and sufficiently responded to my questions. I believe the öansucritp can be accepted for publicatiıon in its current form.

Author Response

We than the reviewer for the positive feedback.

Reviewer 2 Report

Comments and Suggestions for Authors

The author mentions that two figures are provided in the Supplementary Information. However, the author seems to have some problems in the submission process, and I can't find the documents of Supplementary Information in the system. Therefore, I hope the author can resubmit the Supporting Information in the system.

Author Response

We apologize for the inconvenience. We were at that moment convinced that it was uploaded. This time around more attention will be paid to the upload of the SI.